# Surgical Treatment of Corneal Shield Ulcer in Vernal Keratoconjunctivitis: A Systematic Review

**DOI:** 10.3390/jpm13071092

**Published:** 2023-07-02

**Authors:** Samim Azizi, Yousif Subhi, Marie Louise Roed Rasmussen

**Affiliations:** 1Department of Ophthalmology, Rigshospitalet, DK-2100 Copenhagen, Denmark; samim.medschool@gmail.com (S.A.); ysubhi@gmail.com (Y.S.); 2Department of Clinical Research, University of Southern Denmark, DK-5230 Odense, Denmark; 3Department of Clinical Medicine, University of Copenhagen, DK-2100 Copenhagen, Denmark

**Keywords:** shield ulcer, surgery, vernal keratoconjunctivitis, cornea, systematic review

## Abstract

Background: Vernal keratoconjunctivitis (VKC) severely affects the quality of life of affected patients. The development of a shield ulcer is considered one of the most severe late-stage complications, which when untreated leads to irreversible vision loss. In this systematic review, we outlined the results of surgical treatments of corneal shield ulcers in VKC. Methods: We searched 12 literature databases on 3 April 2023 for studies of patients with VKC in which shield ulcers were treated by any surgical treatment. Treatment results were reviewed qualitatively. Assessments of the risk of bias of individual studies were made using the Clinical Appraisal Skills Programme. Results: Ten studies with 398 patients with VKC were eligible for the qualitative review. Two categories of surgical approaches were described: supratarsal corticosteroid injection and debridement with or without amniotic membrane transplantation. Almost all patients experienced resolution or improvement of their shield ulcers, regardless of treatment modality. Time to healing was faster with surgical debridement. A small proportion experienced recurrence and side effects. Conclusions: Surgical treatment for shield ulcers in VKC seems highly effective, but careful post-operative treatment and follow-ups are necessary due to the risk of recurrence and potential side effects.

## 1. Introduction

Vernal keratoconjunctivitis (VKC) is a seasonal determined allergic eye disease characterized by hypersensitivity reactions type 1 and 4 [1]. VKC typically occurs in primary school age and predominantly among males, and often resolves after puberty [1,2]. The prevalence of VKC exhibits ethnical differences, as it is more prevalent in Asia, Central and Western Africa, the Middle East, the Mediterranean, and South America than in Western Europe [1,2]. Although treatment and management of VKC has improved over time [3,4,5], severe and complicated cases remain difficult to manage in clinical practice.

Clinically, VKC is typically classified according to its primary area of affection, i.e., tarsal, limbal, or mixed [6]. In tarsal VKC, the tarsal conjunctiva is classically altered, with the presence of chemosis, hyperemia, and giant papillae. In limbal VKC, Horner–Trantas dots can be seen in the corneal limbus as well as punctate keratitis and corneal shield ulcer [2]. The shield ulcer is a corneal wound considered to be the consequence of the combination of a weakened corneal epithelium and the mechanical friction from conjunctival large papillae [1,2]. The patients experience severe symptoms, including pain and photophobia. An untreated shield ulcer leads to increased risk of keratitis, causes corneal scarring, which can lead to severe astigmatism, amblyopia, corneal neovascularization, and in very severe cases, corneal ulceration and perforation. Thus, shield ulcers can lead to blindness and constitute a significant threat to the patients’ quality of life [1,7]. The incidence of shield ulcers in VKC may vary between countries and through different times of the year, with the highest occurrence typically observed in the spring and summer seasons. Shield ulcer has been classified by Cameron according to three categories depending on its severity: grade 1, which has a transparent base; grade 2, which has a translucent base with or without opaque white or yellow deposits; and grade 3, which has an elevated plaque formation [8].

In this systematic review, we evaluated the published literature on the efficacy of surgical treatments for shield ulcers in VKC. Surgical treatments were stratified according to general themes and outcomes with a special emphasis on the remission and recurrence of shield ulcer.

## 2. Materials and Methods

### 2.1. Study Design

This was a systematic review of clinical studies of patients with VKC with shield ulcers who underwent surgical treatment for their shield ulcers. Institutional board review approval is not relevant for systematic reviews, according to Danish law. We followed the recommendations of the Preferred Reporting Items for Systematic Reviews and Meta-Analyses (PRISMA) [9].

### 2.2. Eligibility of Studies for Review

We defined eligible studies as any clinical study of patients with VKC who underwent any surgical treatment for their shield ulcers. Apart from single case studies, we included all retrospective and prospective studies with patients considered to be representative of the general VKC population and sampled them in a relevant and ideally consecutive manner. Any surgical intervention, including any injection therapy, was considered for this review. We only included studies in the English language, for practical purposes. Animal studies, conference abstracts, and publications without original data were not considered relevant for this review.

### 2.3. Literature Search Strategy and Study Selection

One trained author (Y.S.) searched 12 literature databases on 3 April 2023: PubMed, EMBASE, Cochrane Central, Web of Science Core Collection, BIOSIS Previews, Current Contents Connect, Data Citation Index, Derwent Innovations Index, KCI-Korean Journal Database, Preprint Citation Index, SciELO Citation Index, and ClinicalTrials.gov. No date restrictions were employed. Details of the literature search for each database are provided in Appendix A. 

Two authors (S.A. and Y.S.) screened titles and abstracts of all records to remove records that were duplicates or obviously irrelevant in the context of this review. All remaining references were retrieved as full-text articles to evaluate their potential eligibly for inclusion in the review. The reference lists of these full-text articles were also evaluated for further potentially eligible studies. Two authors (S.A. and M.L.R.R.) evaluated the eligibility of all full-text articles and discussed any disagreements with a third author (Y.S.) to reach a final consensus.

### 2.4. Data Collection, Outcomes of Interest, Risk of Bias of Individual Studies, and Synthesis

Data regarding study design, study characteristics, population details, treatment employed, and clinical outcomes were extracted from each study using data extraction forms. The main clinical outcome of interest was the clinical recurrence of the shield ulcer after treatment. The risk of bias of individual studies was evaluated using the Clinical Appraisal Skills Programme (CASP) Checklist for Cohort Studies [10]. Two authors (Y.S. and M.L.R.R.) performed data extraction and risk of bias assessment. If consensus could not be reached through the methods discussed, a third author (S.A.) was involved to reach a final consensus. 

## 3. Results

### 3.1. Search Results and Study Selection

The literature search identified a total of 268 records from all 12 literature databases. Five records were known to us a priori, and these were added to the pool. Of the total of 273 records, 101 were duplicates, and 159 were obviously irrelevant (e.g., title or abstract clearly irrelevant for VKC or the treatment of shield ulcer). The remaining 13 records were evaluated in full-text form. Of these, 10 full-text articles were deemed eligible for our systematic review [11,12,13,14,15,16,17,18,19,20]. Figure 1 depicts the study selection flow diagram.

### 3.2. General Study Characteristics

The 10 eligible studies included in this review had a total of 398 patients with VKC. Study populations originated from India (*n* = 7), Italy (*n* = 1), Pakistan (*n* = 1), and the United States of America (*n* = 1). Studies were either cases followed in a prospective or retrospective cohort design (*n* = 6) or randomized clinical trials (*n* = 4). The sizes of the populations in the individual studies were: median 19, interquartile range 8–47, range 4–163. Patients were predominantly pediatric (defined as below 18 years of age). In all studies, patients were either predominantly males or exclusively males. Surgical treatment was largely categorized into surgical intervention with corticosteroid injections and surgical intervention with debridement. Further details of studies included in this review, as well as details of the shield ulcers included, are provided in Table 1.

### 3.3. Results after Surgical Intervention with Corticosteroid Injections

Six studies evaluated results after surgical intervention with corticosteroid injections from a total of 142 patients. Follow-up was available for at least 4 months and up to 48 months. Details of treatment and outcomes from these studies are briefly summarized in Table 2.

Anand et al. compared supratarsal injection of triamcinolone (20 mg) with 0.1% cyclosporine/triamcinolone and topical difluprednate [11]. Seventy-eight patients were stratified into three groups with twenty-six patients in each. Supratarsal injection of triamcinolone led to the best outcome of the three groups, with resolution of the shield ulcer occurring in 97% of patients [11].

Aslam et al. examined the efficacy of supratarsal injection of triamcinolone in severe VKC. Eighteen patients were included in the study and received 20 mg triamcinolone each [12]. Resolution of shield ulcer was observed in 20% within one to three weeks after commencement of treatment [12].

Two studies compared the efficacy of supratarsal injection with either dexamethasone or triamcinolone. Horsclaw et al. compared the outcomes after supratarsal injection of either dexamethasone (4 mg/mL) or triamcinolone (40 mg/mL) in 12 patients [14]. This study found no significant difference in the resolution of shield ulcer, which led to the discussion of considering the shorter-acting dexamethasone due to its more favorable overall safety profile [14]. Saini et al. compared the outcomes after supratarsal injection with either dexamethasone (2 mg) or triamcinolone (20 mg) in 19 patients [17]. All cases of shield ulcer were healed, and no differences were found between the groups [17].

Two other studies compared the efficacy of supratarsal injection with either dexamethasone, triamcinolone, or hydrocortisone. Kumar et al. compared the outcomes after supratarsal injection with either dexamethasone (2 mg), triamcinolone (10.5 mg), or hydrocortisone (50 mg) in 48 patients [15]. However, only 8 had shield ulcer, and at 3 weeks all cases of shield ulcer were healed [15]. Singh et al. compared the outcomes after supratarsal injection with either dexamethasone (2 mg), triamcinolone (10.5 mg), or hydrocortisone (50 mg) [19]. Resolution of shield ulcer was observed after 3 weeks for all cases regardless of treatment [19]. 

### 3.4. Results after Surgical Intervention with Debridement

Six studies evaluated results after surgical intervention with debridement from a total of 171 patients. Follow-up was available for at least 2–25 months in studies that reported follow-up periods. Details of treatment and outcomes from these studies are briefly summarized in Table 3.

Caputo et al. performed surgical debridement on four patients with shield ulcers [13]. Ulcer plaques were scraped using a crescent knife and the epithelium was removed up to 1 mm around the plaque, leaving the surface of the corneal stroma as smooth as possible [13]. Patients received cyclosporine treatment after surgery until complete remission, which was achieved within 4–5 days [13].

Reddy et al. treated 163 patients (193 eyes) with shield ulcers with either medical treatment or by performing surgical debridement [16]. In cases with surgical debridement, each ulcer was scraped at its base and in its margin, and in patients who did not show signs of re-epithelization within 2 weeks, an amniotic membrane transplantation was performed [16]. During a relatively long follow-up period of 544 ± 751 days, 118 eyes (61%) were managed medically, and 75 eyes (39%) underwent surgical debridement, of which 44 also had amniotic membrane transplantation [16]. This strategy led to an overall rate of resolution of 94%, with 15% experiencing recurrence during the follow-up period [16].

Sharma et al. performed a modified surgical technique of continuous intraoperative optical coherence tomography-guided shield ulcer debridement and amniotic membrane transplantation on four patients [18]. The benefit of this technique is described as being able to visualize any residual plaque as a hyperreflective membrane and dots, thus being able to obtain real-time feedback to ensure the complete removal of the ulcer [18]. Resolutions were obtained within 7–12 days [18].

Sridhar et al. performed surgical debridement and amniotic membrane transplantation in four patients, of which two also received supratarsal injection with corticosteroids [20]. The plaque was removed, the ulcer bed was thoroughly debrided, and the amniotic membrane was positioned over the epithelial defect [20]. All cases experienced resolution within 2 weeks [20].

### 3.5. Risk of Bias of Individual Studies

Risk of bias of individual studies showed that studies generally addressed a focused issue, recruited in an acceptable way, and accurately described the treatments employed (exposure). Measurement of the outcome when using definitions of shield ulcer was unclear in all studies apart from two. Identification and addressing of potential confounding factors were only performed in four studies, which were all randomized clinical trials. An adequate follow-up period after surgical treatment of shield ulcer was considered to be at least three months, and this was present in all but two studies. Details of the evaluation of the risk of bias of individual studies are available as Table 4. 

## 4. Discussion

Our systematic review highlights the overall beneficial results from both supratarsal injection with corticosteroids and surgical debridement as nearly all obtain remission. However, findings of the studies and the necessity of intense postoperative eyedrop regimens and controls also highlight that the management of this severe type of VKC can be complicated. 

The mechanism of action of supratarsal corticosteroid injection relies on a general suppression of the local immune system. The rapid reduction in giant papilla formation helps prevent direct mechanical trauma, while deactivating the toxic major basic proteins released by activated eosinophils within the shield ulcer debris, thereby creating a favorable environment for corneal epithelial healing [16]. Studies in this review found that corticosteroid injections have high efficacy in the treatment of shield ulcer. The average time required for healing across the studies was consistently within the range of 2–3 weeks. Interestingly, the specific type of corticosteroid used did not seem to significantly affect the healing of shield ulcers. This contrasts with the general knowledge that specific corticosteroids have different potencies [21]. Relative to each other, a ranking order of potency (from highest to lowest) is dexamethasone, triamcinolone, and hydrocortisone [21]. However, these potency-related pharmacological aspects, including side effects, such as systemic absorption with the concern of potential adrenal insufficiency and inhibition of growth hormones, remain poorly understood when given supratarsal or topical on the conjunctiva [22]. Ocular side effects of corticosteroid injections are generally well-described and also seen in studies of this review. In two studies [11,14], 6–8% of patients experienced elevated intraocular pressure (IOP) for months following triamcinolone injections.

Recurrence of the shield ulcer may be influenced by the type of glucocorticoid used [17]. This was demonstrated by Saini et al., in which both dexamethasone and triamcinolone injections led to complete resolution of shield ulcer; however, the recurrence of shield ulcer occurred after 6–20 days after dexamethasone treatment, whereas the recurrence occurred after 180–290 days after triamcinolone treatment [17]. This difference highlights the known pharmacodynamic differences between dexamethasone and triamcinolone, i.e., triamcinolone is more long-acting [21]. 

Surgical debridement led to faster healing, either with or without amniotic membrane transplantation. Time to healing ranged within 4–17 days. Since debridement involves the removal of the dense plaque tissue, one can speculate about the removal of this plaque, which is likely composed of cytotoxic eosinophilic granule major basic proteins [16,23]; removal of these proteins from the local milieu can hasten the healing and re-epithelialization process.

Among the included studies, one study exclusively performed surgical debridement [13]. The remaining three studies [16,18,20] utilized a combination of debridement, amniotic membrane transplantation, and supratarsal corticosteroid injections. Two studies employed a stepwise approach, with the most severe cases of grade 3 shield ulcer receiving debridement, amniotic membrane transplantation, and supratarsal corticosteroid injections using dexamethasone [16,20]. The post-operative treatment varied among the studies but generally involved a combination of the following: topical antibiotic treatment for the initial weeks, often with fluoroquinolones or moxifloxacin; topical corticosteroids, topical cyclosporine, and lubricating eyedrops, and topical cromoglycate. These treatment modalities aimed to address the shield ulcer and to prevent its recurrence [13,16,18,20]. However, despite intense topical postsurgical treatment, Reddy et al. reported recurrence rates of 13–14% in a large Indian population, even with a stepwise treatment approach [16]. Complications reported in the studies include secondary bacterial keratitis (10%), disintegration of the amniotic membrane, and corneal scarring [16,20]. It is important to note that in the context of a severe shield ulcer of grade 3, corneal scarring may be considered a natural progression rather than a distinct complication.

Limitations of this review should be noted when interpreting its results. Eight of ten studies in this review are from India and Pakistan, which challenges the generalizability of results to other populations. However, this also reflects the epidemiology of the disease, as VKC and shield ulcer are more commonly observed in tropical regions such as India, Pakistan, central Africa, and certain Middle Eastern countries. The specific factors responsible for this epidemiological phenomenon are not fully understood; however, a hot and humid climate or increased exposure to airborne allergens are speculated to be contributing factors [24]. Another important limitation is that none of the studies had a placebo or an observation group, which makes it difficult to estimate the efficacy of these treatment modalities. However, considering that patients are severely symptomatic and the reports of the natural history of shield ulcer without treatment, it would be ethically challenging to design a study with a group that does not receive the standard of care. Further studies on the treatment of shield ulcers are warranted.

It should also be noted that there are important adjacent aspects of shield ulcer management that are not included in this review. First, the use of cyclosporine and tacrolimus before and after surgery may facilitate the recovery process [25]. In general, for VKC, and especially in the presence of any corneal complication, the patient and the parents should be advised to achieve control of ocular rubbing, as it plays an important role in the etiology of progression. Finally, from a surgical perspective, the use of intraoperative anterior segment OCT may allow for a more precise depth of debridement and placement of an amniotic graft [18].

In conclusion, we here report the efficacy of surgical treatments of corneal shield ulcers in VKC. Overall, two major approaches are available: supratarsal corticosteroid injection, and debridement with or without amniotic membrane transplantation. The utilization of the Cameron grading system to assess the severity of shield ulcers provides a reasonable method to distinguish between shield ulcers in a clinically meaningful manner. Using the Cameron grading system, our review suggests that grade 1 shield ulcers can be treated using topical steroids or supratarsal injection of corticosteroids, whereas grade 2 and 3 shield ulcers may need debridement +/- supratarsal injection of corticosteroids as well as amniotic membrane transplantation over the debridement in cases with deep stromal involvement to increase the speed of healing.

## Figures and Tables

**Figure 1 jpm-13-01092-f001:**
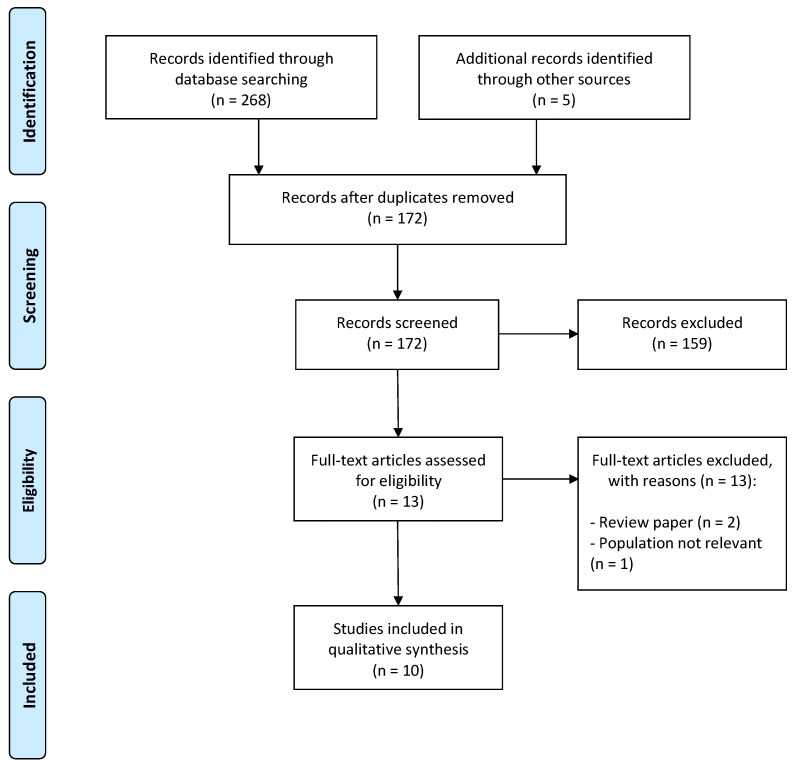
The Preferred Reporting Items for Systematic Reviews and Meta-Analyses (PRISMA) flow chart for study selection.

**Table 1 jpm-13-01092-t001:** Study and population characteristics.

Reference	Country	Study Design	Patients, N	Age, Years, Range	Gender, Males	Inclusion Criteria	Exclusion Criteria
Anand et al. (2017) [11]	India	Randomized clinical trial	78	7–21	65%	Severe refractory VKC, with symptoms interfering with daily life. Cobblestone papillae, Horner–Trantas dots, and corneal involvement (shield ulcer, limbal stem cell deficiency, punctate keratitis, limbal pannus).	Any VKC less severe, having inflammation-free interval of greater than 2–3 months per year, raised intraocular pressure (IOP), patients on anti-glaucoma medications
Aslam et al. (2017) [12]	Pakistan	Prospective cohort	18	5–25	72%	Severe refractory VKC, all treated with natrium cromoglycate 4% × 4 daily, lodoxamide 0.1%, prednisolone 0.1% without result for at least 4 months.	Patient with systemic disease, history of ocular surgery, less than 4 months of follow-up
Caputo et al. (2012) [13]	Italy	Prospective cohort	4	6–14	75%	VKC, ongoing treatment with cyclosporine A, presence of shield ulcer with elevated plaques (grade 3) unresponsive of medical treatment for at least 1 month.	Less than 12 months of follow-up, patients receiving systemic therapy for at least one month
Horsclaw et al. (1996) [14]	USA	Prospective cohort	12	9–28	67%	Severe refractory VKC with symptoms; despite stepwise treatment with preservative artificial tears, topical sodium cromolyn 4%, lodoxamide tromethamine 0.1%, ketorolac tromethamine 0.5%, prednisolone acetate 0.125, prednisolone acetate 1% and antibiotic if needed.	Topical cyclosporine or systemic treatment
Kumar et al. (2022) [15]	India	Randomized clinical trial	48	5–25	85%	Severe refractory VKC non-responders to a monthlong maximum topical therapy.	Active ocular infection, patients who were concurrently treated for other allergic disorders
Reddy et al. (2013) [16]	India	Retrospective cohort	163	12–14	89%	Severe VKC with shield ulcer in grade 1–3.	Patients with a previous history of any corneal surgery, patients not compliant with medications, patients who did not comply with follow-ups at regular weekly intervals until re-epithelialization, and patients with untreated concurrent problems that would affect re-epithelialization
Saini et al. (1999) [17]	India	Randomized clinical trial	19	9–23	100%	Severe refractory VKC with corneal complications (punctate keratitis, shield ulcer, pannus) and ongoing symptoms within minimum > 6 weeks.	Patients with history of contact lens wear and unable to communicate because of age or intellect
Sharma et al. (2018) [18]	India	Prospective cohort	7	6–10	71%	VKC patient with shield ulcer with plaque planned to undergo shield ulcer debridement.	None mentioned
Singh et al. (2001) [19]	India	Randomized clinical trial	45	5–23	89%	Advanced and refractory VKC not responding or inadequately responding to a monthlong maximum topical therapy with NSAID, mast cell stabilizer, antihistamine, and corticosteroids.	Active ocular infection and patients concurrently treated for other allergic disorders
Sridhar et al. (2001) [20]	India	Retrospective cohort	4	6–18	100%	Severe VKC with shield ulcer of grade 2–3.	None mentioned

Abbreviations: USA = United States of America; VKC = vernal keratoconjunctivitis.

**Table 2 jpm-13-01092-t002:** Studies of corticosteroid injections for shield ulcer in vernal keratoconjunctivitis (VKC).

References	Treatment	Follow-Up	Postoperative Eyedrops	Postoperative Complications	Results on Shield Ulcer	Recurrence of Shield Ulcer
Anand et al. (2017) [11]	Three groups. Group A: topical difluprednate with tapering for 1 month; Group B: triamcinolone injection supratarsal 0.4 mL (20 mg); Group C: topical cyclosporin/tacrolimus 0.1% for 2 months	9 months	Antibiotics one week after triamcinolone in group B.	Group B: elevated IOP in 6%, redness after injection in 9%	After one month: Group A: shield ulcer resolved in 69%; Group B: shield ulcer resolved in 97%; Group C: shield ulcer resolved in 0%	After 3 months: Group A + B: 20–30%; Group C: 50–60%
Aslam et al. (2017) [12]	Supratarsal triamcinolone acetonide injection 0.5 mL (20 mg)	4–48 months	Natrium cromoglycate 4% × 4 daily	No complications	Resolution of shield ulcer in 20% within 1–3 weeks. Resolution in all at end of follow-up.	None during follow-up
Horsclaw et al. (1996) [14]	Supratarsal injection of corticosteroid, 2 groups. One group in supratarsal dexamethasone (short), another group in triamcinolone (long acting).	4–48 months	Topical cromolyn sodium 4% and ketorolac tromethamine 0.5% × 4 daily, if shield ulcer prophylactic topical ciprofloxacin hydrochloride	One patient with persistent IOP elevation for months, treated with levobunolol 0.5% × 2 daily	Resolution of shield ulcer in 16 days with dexamethsaone and 14 days with triamcinolone	None during follow-up
Kumar et al. (2022) [15]	Supratarsal injection of corticosteroids, 3 groups. Group A: 2 mg dexamethasone; Group B: 10.5 mg triamcinolone, Group C: 50 mg hydrocortisone.	6 months	Cromoglycate 4% × 4 daily and cold compresses	No complications	Resolution of shield ulcer healed in 3 weeks in all eyes. No differences between the three groups.	None during follow-up
Saini et al. (1999) [17]	Dexamethasone (2 mg) vs. triamcinolone (20 mg) injection in one eye of each patient.	12 months	Sodium cromoglycate 4% × 2 daily + diclofenac sodium 0.1% × 4 daily	No complications	Resolution of shield ulcer in 14–16 days.	In 4 eyes after 6–20 days after dexamethasone treatment, and in 3 eyes after triamcinolone treatment after 180–290 days.
Singh et al. (2001) [18]	Supratarsal injection of corticosteroids, 3 groups. Group A: 2 mg dexamethasone; Group B: 10.5 mg triamcinolone, Group C: 50 mg hydrocortisone. Three weeks between each eye. Initially two weeks of wash-out of other treatments with disodium cromoglycate 2% × 4 daily	6 months	Disodium cromoglycate 2% eye drops × 4 daily	No complications	Resolution of shield ulcer in 3 weeks in all patients in all groups.	None during follow-up

Abbreviations: IOP = intraocular pressure; VKC = vernal keratoconjunctivitis.

**Table 3 jpm-13-01092-t003:** Studies of corticosteroid injections for shield ulcer in vernal keratoconjunctivitis (VKC).

References	Treatment	Follow-Up	Postoperative Eyedrops	Postoperative Complications	Results on Shield Ulcer	Recurrence of Shield Ulcer
Caputo et al. (2012) [13]	Surgical debridement	18–24 months	Topical fluoroquinolone × 2 daily for 1week, preservative-free lubricating eye drops, cyclosporine A 1% × 4 daily.	No complications mentioned	Resolution within 4–5 days.	No recurrence.
Reddy et al. (2013) [16]	Stepwise treatment. Step 1: medical treatment; step 2 + 3: medical treatment + debridement +/-amnion membrane transplantation	18–25 months	Sodium cromoglycate 2% or 4% × 2 daily; topical corticosteroids: prednisolone acetate 1% or fluorometholone ophthalmic suspension 0.25%; antibiotic eye drops × 4 daily; and lubricating eye drops × 6–8 daily.	Secondary bacterial keratitis	Resolution in 94% in mean 17 days.	Recurrence in 15%.
Sharma et al. (2018) [18]	Intraoperative OCT guided surgical debridement + steroid therapy + amnion membrane transplantation	2 months	Topical olopatadine hydrochloride 0.1% × 2 daily, topical prednisolone acetate 1% × 6 daily, moxifloxacin × 3 daily, lubricating eye drops × 8 daily.	No complications mentioned	Resolution within 7–12 days.	No recurrence.
Sridhar et al. (2001) [20]	Surgical debridement + amnion membrane transplantation. Two patients were also treated with steroid injection therapy	Unknown	Topical steroid in all patients; three patients also received sodium cromoglycate; one patient also received cyclosporine.	Disintegration of the amnion membrane transplant, corneal scarring	Resolution within 2 weeks.	One recurrence after debridement and amnion membrane transplantation.

Abbreviations: OCT = optical coherence tomography.

**Table 4 jpm-13-01092-t004:** Risk of bias of individual studies using the Clinical Appraisal Skills Programme (CASP) Checklist.

Reference	Q1	Q2	Q3	Q4	Q5	Q6
Anand et al. (2017) [11]	Yes	Yes	Yes	Unclear	Yes	Yes
Aslam et al. (2017) [12]	Yes	Yes	Yes	Unclear	No	Yes
Caputo et al. (2012) [13]	Yes	Yes	Yes	Yes	No	Yes
Horsclaw et al. (1996) [14]	Yes	Yes	Yes	Unclear	No	Yes
Kumar et al. (2022) [15]	Yes	Yes	Yes	Unclear	Yes	Yes
Reddy et al. (2013) [16]	Yes	Yes	Yes	Yes	No	Yes
Saini et al. (1999) [17]	Yes	Yes	Yes	Unclear	Yes	Yes
Sharma et al. (2018) [18]	Yes	Yes	Yes	Unclear	No	No
Singh et al. (2001) [19]	Yes	Yes	Yes	Unclear	Yes	Yes
Sridhar et al. (2001) [20]	Yes	Yes	Yes	Unclear	No	No

Questions: Q1: Did the study address a clearly focused issue. Q2: Was the cohort recruited in an acceptable way. Q3: Was the exposure accurately measured to minimize bias. Q4: Was the outcome accurately measured to minimize bias. Q5: Have the authors identified all important confounding factors and have they taken account of the confounding factors in the design and/or analysis. Q6: Was the follow-up of subjects complete enough and was the follow-up of subjects long enough. For every question, one states “Yes”, “Unclear”, or “No”.

## Data Availability

No new data were created or analyzed in this study. Data sharing is not applicable to this article.

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
