# Peer review of "Surgical Treatment of Corneal Shield Ulcer in Vernal Keratoconjunctivitis: A Systematic Review"

_jpm, 2023, doi:10.3390/jpm13071092_

Round 1

Reviewer 1 Report

Dear Authors

You have done a very interesting and methodologically well developed work.

Good tool the program you use and cite as reference 10.

A semantic detail, in the results: line 98.

Explain why "obviously" irrelevant, those 159 papers. Or describe in methods, what they refer to as irrelevant. It is that "obviously" is not something precise. Clarify this point.

The rest of the presentation of results is fine. Table 4 is interesting, in this type of work. It is something else. Very good.

The discussion is enriching and arrives at a conclusion consistent with the results obtained, but at the same time, with a practical message in the clinical setting, to be able to take a therapeutic measure in relation to Cameron's grading system.

I believe that their work will be of great utility.

Just please clarify "obviously irrelevant" so that your work is solid in that aspect and perhaps minor aspects of wording. 

Congratulations.

Line 64. 67 67, search synonyms to avoid repeating the word consider/ing

Author Response

Thank you for your time and comments. We agree on all points. The term “obviously irrelevant” is the preferred phrasing as outlined in the Cochrane Handbook ( https://training.cochrane.org/handbook/current/chapter-04 ). To accommodate to the concerns of the reviewer, we have now provided a clarification in the manuscript as recommended.   

Reviewer 2 Report

The authors addressed a very interesting, up-to date field in ophthalmology - Surgical Treatment of Corneal Shield Ulcer in Vernal Keratoconjunctivitis: A Systematic Review. The main body of the paper is very well written and clearly structured. A few spelling mistakes. Otherwise, I would suggest to accept the article for publication as presented.

Author Response

Thank you for your time and comments. We have reviewed and corrected the few spelling errors. 

Reviewer 3 Report

Shield ulcer is a shallow indolent ulcer associated with vernal keratoconjunctivitis (VKC) presenting with severe photophobia, whitish opacity in corneadiminished vision, and red itchy eyes. The number of reported cases in the western world is small because the disease is rare. Therefore, such a systematic study with large number of cases is necessary for the clinicians to prove the effectiveness of the described methods. The current study is an addition to the literature as it follows standard protocols. The review is very comprehensive. I have to add very minor comments that would increase the importance of the paper view my large experience in that field.

The use of intraoperative anterior segment OCT ensures better placement of amniotic graft and correct depth of the debridement.

The use of cyclosporine and tacrolimus throughout before and after surgery is crucial for faster recovery.

Rubbing is crucial in the etiology of the disease and controlling ocular rubbing allows faster healing and no recurrences.

Topical tacrolimus for allergic eye diseases.

Erdinest N, Ben-Eli H, Solomon A.Curr Opin Allergy Clin Immunol. 2019 Oct;19(5):535-543. doi: 10.1097/ACI.0000000000000560

Author Response

Thank you for your time and comments. We completely agree and have now highlighted these points in the discussion section of our manuscript.